# Crossover from positive to negative optical torque in mesoscale optical matter

Fei Han [1], John A. Parker[2,3], Yuval Yifat [2], Curtis Peterson[2,4], Stephen K. Gray[5], Norbert F. Scherer[2,4] & Zijie Yan [1]

The photons in circularly polarized light can transfer their quantized spin angular momentum to micro- and nanostructures via absorption and scattering. This normally exerts positive torque on the objects wher the sign (i.e., handedness or angular direction) follows that of the spin angular momentum. Here we show that the sign of the optical torque can be negative in mesoscopic optical matter arrays of metal nanoparticles (NPs) assembled in circularly polarized optical traps. Crossover from positive to negative optical torque, which occurs for arrays with different number, separation and configuration of the constituent particles, is shown to result from many-body interactions as clarified by electrodynamics simulations. Our results establish that both positive and negative optical torque can be readily realized and controlled in optical matter arrays. This property and reconfigurability of the arrays makes possible programmable materials for optomechanical, microrheological and biological applications.

[1] Department of Chemical and Biomolecular Engineering, Clarkson University, Potsdam, NY 13699, USA. [2] James Franck Institute, The University of Chicago, Chicago, IL 60637, USA. [3] Department of Physics, The University of Chicago, Chicago, IL 60637, USA. [4] Department of Chemistry, The University of Chicago, Chicago, IL 60637, USA. [5] Center for Nanoscale Materials, Argonne National Laboratory, Lemont, IL 60439, USA. Correspondence and requests for materials should be addressed to Z.Y. (email: zyan@clarkson.edu)

The interaction of light with matter is manifest in an object's response to incident light due to momentum transfer from photons to matter[1,2], which normally leads to movement of the object concomitant with the photon momentum. For example, an optical beam with linear momentum creates positive optical force (i.e., radiation pressure[1]) on an object leading to its forward movement. Similarly, light can spin or rotate a microscopic object if it carries angular momentum. The transfer of spin angular momentum from photons to matter normally exerts positive torque that rotates the object along the direction of the electric vector of light[2–6]. However, momentum transfer can also cause exotic dynamics of objects that defy our intuition[7,8]. Negative optical force can arise with specific combinations of specially designed or configured optical beams and particles[7,9–12], where backwards motion occurs concomitantly with preferential light scattering in the forward direction.

Negative optical torque, which causes objects to rotate opposite to the direction of the incident beam's angular momentum, has been theoretically postulated[13–15] yet with limited experimental realization[8,16]. Stable negative optical torque has only been demonstrated via coupled spin–orbit light scattering in optically inhomogeneous and anisotropic transparent media, such as birefringent and structured glass disks[8] or polymer layers[16]. Negative optical torque events were also observed in electro-dynamically coupled NP dimers[17], but only when the dimers were in a transient (unstable) state. Furthermore, the associated theory these authors presented was not formulated to predict negative optical torque for three or more NPs stabilized by optical binding forces[18–20]. It remains a fundamental challenge to realize and explain stable and tunable negative optical torque in a material system with robust and persistent orbital motion.

We demonstrate here that negative optical torque can be generated in mesoscale optical matter[21] consisting of self-organized silver NPs in circularly polarized optical beams. Moreover, the self-organization aspect of the optical matter arrays reveals a new approach to control and switch optical torque in structured nanomaterials. The negative optical torque arises without resorting to explicitly birefringent materials or inhomogeneous media. Instead, achieving negative optical torque relies on the number, separation and configuration of constituent NPs. Our experimental results clearly show a crossover from positive to negative optical torque as a function of the number of NPs in the optical matter arrays and for various configurations.

Our theory and simulations reveal that the wavelength of light, the geometry and symmetry, interparticle separation, and induced dipole interactions of the optical matter arrays play important roles in the emergence of negative optical torque, including a close connection with the lattice plasmon (LP) modes[22,23] supported by extended optical matter arrays.

## Results

**Mesoscale optical matter arrays**. Figure 1a illustrates the experimental setup where a circularly polarized Gaussian beam that has been truncated by an iris at a prior conjugate plane is focused to the back aperture of an objective, creating a collimated beam at the surface of a coverslip where NPs are trapped (Methods section). The optical trap's intensity profile is designed to be relatively flat in order to minimize the influence of radially compressive gradient forces[19,24,25]. Light-mediated self-organization of silver NPs leads to the formation of mesoscale optical matter arrays. Figure 1b shows a series of representative structures with NPs occupying sites in a hexagonal lattice. The optical matter isomers (i.e., arrays with the same number of NPs) may have structural transitions, but some structures (defined as type-I) appear more often than others (types II and III are shown in Fig. 1b and more structures can be seen in Supplementary Note 1), so our analyses mainly focus on the type-I structures. We find that an optical matter structure is more likely to form when it has more nearest neighbor pairs with (shorter) distances that enhance the mutual interaction associated with optical binding[20]. In Supplementary Note 1, we estimate the effective potential energy barrier for single NP jumps between isomers to be $\sim 6k_BT$ in the 2-NP, 7-NP, and 10-NP arrays. That value is larger than the thermal energy of the NPs thus enabling optical binding, but not high enough (e.g., $10k_BT$) to obviate the efficacy of stochastic kicks in the Brownian motion[26] that will change the potential energy landscape and lead to various configurations (i.e., isomers)[27].

**Crossover from positive to negative optical torque**. Our key results are shown in Fig. 2; notably that the sign of rotational motion of an optical matter array, and thus its optical torque, depends on the number and configuration of the constituent particles (see Supplementary Movies 1 and 2). For example, a 3-I trimer rotates counterclockwise in the direction of our LHCP

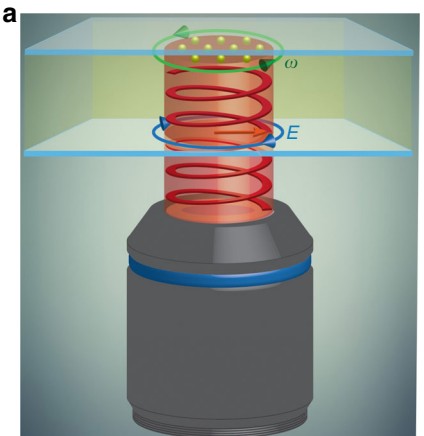
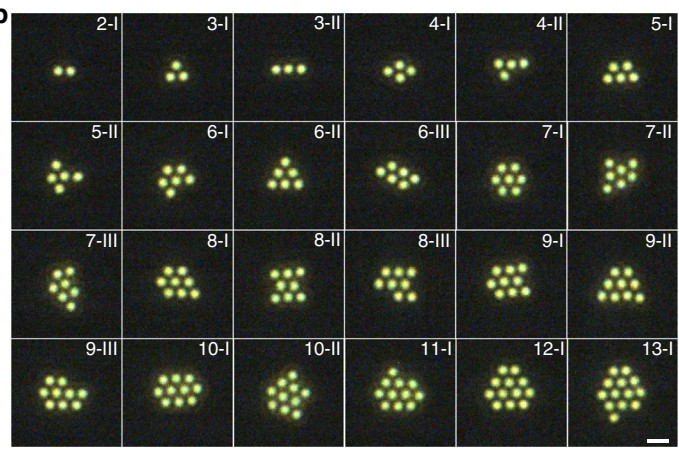

**Fig. 1** Light-driven self-organization of silver NPs into mesoscale optical matter arrays. **a** Schematic of rotation of an optical matter array in a liquid sample cell illuminated by a circularly polarized laser beam. Blue arrow represents the rotation direction of the electric vector of light, and green arrow indicates the rotation direction of an array formed by optical binding of ten NPs. The NPs are experimentally observed from the point of view of the incident light source, where the electric vector rotates counterclockwise in the imaging plane. We use the convention that this is termed left-handed circularly polarized (LHCP). **b** Dark-field optical images of representative optical matter arrays consisting of 2–13 150 nm diameter silver NPs. The scale bar is 1 μm

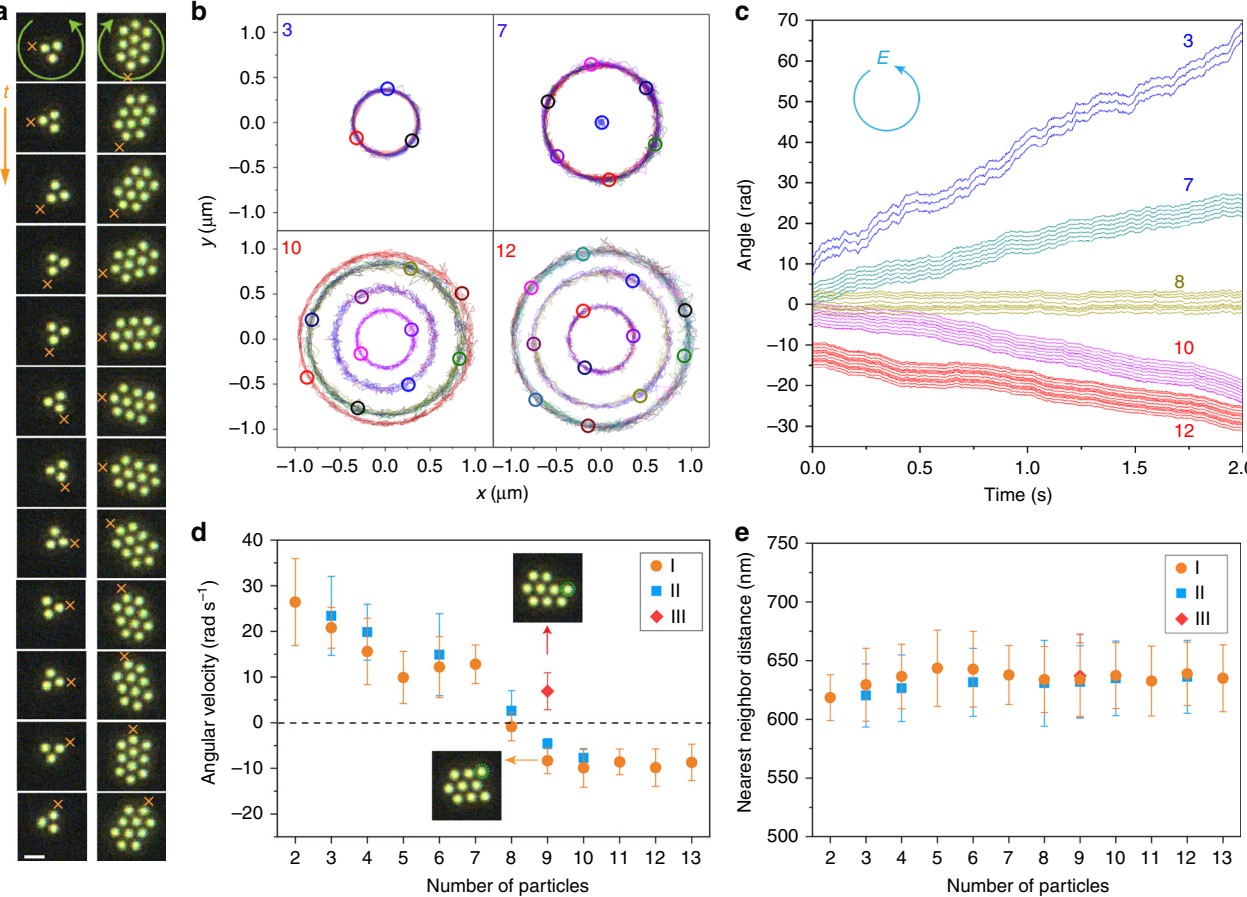

**Fig. 2** Experimental determination of optical torque crossover as a function of optical matter array size. **a** Sequences of optical images of a 3-I array and a 10-I array in a LHCP laser beam. The time interval between frames is 0.01 s for the 3-I array and 0.033 s for the 10-I array. The two arrays clearly exhibit opposite rotation directions as indicated by following the × symbols although they are illuminated with the same LHCP optical beam. The scale bar is 1 μm. **b** Trajectories of four different optical matter arrays relative to the center of mass of each array for 2 s. The open circles are representative instantaneous configurations. **c** Time trajectories of orientations of individual NPs in each array relative to its center of mass. The central particle in the 7-I array overlaps with the center of rotation and therefore is not shown. **d** Average angular velocity of optical matter arrays as a function of the number of particles in each array. The colored symbols delineate type-I, II, III structures (see Fig. 1 for examples). The insets are optical images of two special arrays with 9 NPs, where the only difference is the position of one NP-indicated by the green circle. **e** The mean values of all nearest neighbor distances of different arrays. Error bars in **d** and **e** correspond to one standard deviation

beam due to positive torque exerted by the beam, whereas a 10-I array rotates clockwise due to negative torque (Fig. 2a).

The optically bound NP arrays behave as rigid bodies that rotate around their centers of mass, which can be clearly seen from the trajectories of four different arrays plotted in Fig. 2b. Note that negative torque refers to the total orbital torque exerted on the whole array, and therefore one can also view the whole array as spinning. However, due to the discrete nature of the array, we prefer to refer to its motion as rotation to better descibe the orbital dynamics of the NPs. We also expect that the individual NPs are spinning about their individual axes in circularly polarized light, which is not directly observable in optical images but has been demonstrated, for example, by the oscillating autocorrelation function of scattering intensity[6]. Hydrodynamic coupling may occur between two optically bound and spinning particles, where the spin torque can couple to the orbital torque due to the liquid flow created by the spinning particles[14]. In Supplementary Note 2, we discuss possible implications of hydrodynamic coupling in a 7-NP array and conclude that the hydrodynamic spin–orbit coupling would not cause torque torque reversal in optical matter arrays.

The unwrapped angular trajectories (Fig. 2c) have positive slopes, i.e., angular velocities, for the small arrays and negative slopes for larger arrays, while the 8-I array is nearly stationary. Since the incident electric vector rotates counterclockwise in the imaging plane, negative angular velocities of arrays reflect negative optical torques. The angular velocities of a series of optical matter arrays are plotted in Fig. 2d. The crossover from positive to negative velocity for the type-I arrays occurs at 8 particles. Type-II arrays have a similar trend. However, there are interesting anomalies near the crossover value. Arrays 9-I and 9-III show opposite rotation directions even though they have nearly the same distribution of nearest neighbor distances. They switch rotation direction by just shifting the 9th particle by one lattice site. Therefore, optical torque reversal can be induced by structural transitions in optical matter isomers as well as by changing the number of NPs in the array.

The corresponding nearest neighbor distances are shown in Fig. 2e, which are almost constant at around 640 nm. However, Supplementary Note 3 shows an amazing result that when the separations in the arrays are reduced by tuning the electrostatics, the crossover from positive to negative torque can even occur between 2 and 3-NP arrays. In addition, a slight change of the geometry of a 7-NP array can reverse the torque from negative to positive (see Supplementary Movie 3). These results demonstrate that interparticle separation and geometry influence the sign of

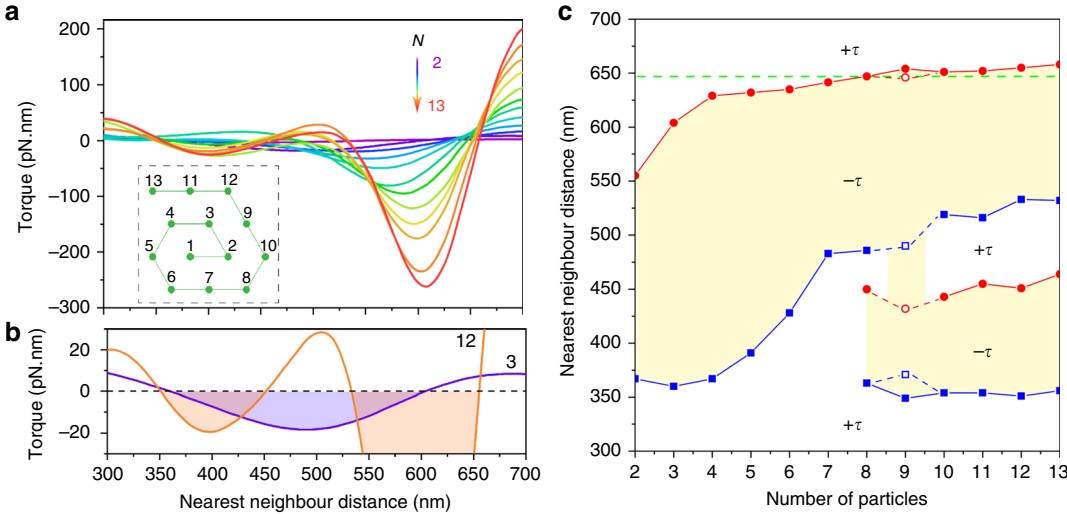

**Fig. 3** FDTD simulation of optical torques in static optical matter arrays composed of ideal hexagonal NP lattices with varying lattice constants. **a** Calculated optical torque in type-I arrays with increasing number of particles (N) and lattice constants. The inset illustrates the configurations of the arrays. **b** Zoom-in view of the torque vs nearest neighbor separation highlighted for the 3-I and 12-I arrays, where the negative torque regions are shaded. **c** An optical torque "phase diagram" where the torque direction of an array depends on its lattice constant (i.e. interparticle spacing) and number of particles. The solid symbols are boundaries separating regions of positive and negative torque, and the open symbols at 9 particles is for a 9-III array. The green dashed horizontal line at 647 nm interparticle separation indicates the torque crossover at 8 particles in reference to Fig. 2

the optical torque on the arrays. Similarly, it was recently found that the separation and orientation of two optically bound particles could impact the negative forces in a tractor-beam[28].

**Optical torque phase diagram**. We used finite-difference time-domain (FDTD) electrodynamics simulations to calculate the optical torque in the type-I arrays to further characterize the optical torque crossover behavior (Methods section). Figure 3 shows that the optical torque for each array oscillates between positive and negative values depending on the lattice constant and number of particles. As a result, the optical torque can be positive for some arrays but negative for others at a certain lattice constant. Figure 3b shows a zoom-in view of the torque reversal behavior for arrays 3-I and 12-I, where the former has one negative region and the latter has two for the 300 to 700 nm range of lattice constants. The lattice constant values for the torque reversals allow creating an optical torque "phase diagram" separating regions of positive and negative torque. These phase diagrams are a function of the lattice constant and number of particles (Fig. 3c). A negative torque region exists for all the arrays, i.e., the upper yellow region. The upper boundary of this region is close to the optical binding distance of Fig. 2 and provides an insight into the torque reversal observed in our experiments. For example, torque reversal for 8 particle arrays occurs at 647 nm nearest neighbor separation (i.e., the green horizontal line). The simulations also show that different isomers can have different torque reversal behaviors; e.g., arrays 9-I vs 9-III in Fig. 3c (see details in Supplementary Note 4). Further simulations show that the coverslip near optical matter arrays (see Fig. 1a) has little influence on the optical torque, and thermophoretic force (induced by laser-heated NPs) is negligible at optical binding separation (Supplementary Notes 5 & 6).

**FDTD-particle dynamics simulations**. The phase diagram reveals a rich parameter space for the optical torque in an optical matter array. However, it does not provide information about the dynamics of arrays. The light-driven self-organization and orbital motions of NPs in circularly polarized optical beams, including the angular velocities, is captured by a FDTD-particle dynamics

simulation approach (Methods section and Supplementary Note 7). Simulations for a range of arrays with neutral NPs are shown in Fig. 4a. The dimer experiences positive optical torque while the tetramer and larger arrays experience negative optical torques, with the trimer as the crossover. The equilibrium separations of these arrays are smaller than the experimental results of Fig. 2, where the interparticle separations are increased by repulsive electrostatic forces due to negatively charged polymer layers on the Ag NPs. By assuming charged NPs in the simulation, the reversal of optical torque shifts to arrays of 8 NPs (Fig. 4b) when the interparticle distance is 640 nm (Fig. 4c). Figure 4d summarizes the optical torque crossover for charged and uncharged type-I arrays. Optical torque values for type-II and III arrays are also plotted, revealing that different isomers can exhibit opposite rotation directions. These results agree with the experimental observations, e.g., 9-I and 9-II rotate clockwise while 9-III rotates counterclockwise. The simulations also predict that by decreasing the interparticle separations, the optical torque may be reversed from positive to negative in optical matter arrays with fewer than 8 NPs. We have experimentally demonstrated that by tuning the interparticle separations by adding positively charged surfactants to the polyvinylpyrrolidone (PVP)-coated Ag NP solution (Supplementary Note 3). We also find that the sign of optical torque depends on the trapping laser wavelength. For example, at $\lambda = 710$ nm, the type-I arrays with 9–11 NPs show positive optical torques (Supplementary Note 8), different from the cases at $\lambda = 800$ nm. In addition, real Ag NPs have facets, yet our simulations using irregular particles give similar results as for smooth spheres (Supplementary Note 9).

**Mie theory calculations**. Achieving a more fundamental understanding of negative optical torque requires simulation of matter-radiation interactions with and without interparticle interactions, which is challenging to do within the FDTD framework. We therefore developed a Generalized Mie Theory (GMT)-Langevin dynamics method (see Methods) and simulated a 7-particle array in a LHCP plane wave without (Fig. 5a) and with (Fig. 5b) induced dipole interactions among the NPs. The NPs are optically bound in both cases and thermally fluctuate ($T = 298$ K) around

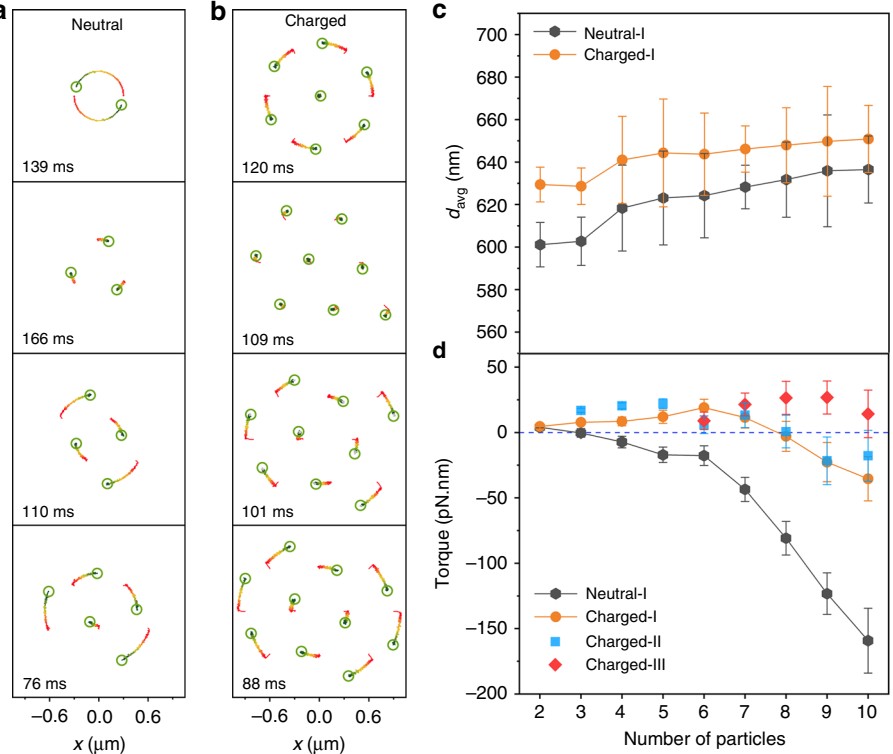

**Fig. 4** FDTD-particle dynamics simulation of light-driven self-organization and rotation of mesoscale optical matter arrays. **a** Rotation of optical matter arrays with neutral NPs governed by optical binding interactions for different numbers of particles in the array. Directions of rotation are indicated by red to green color with open circles showing the final positions of the NPs. The duration of the trajectory is labeled in each panel. **b** Rotation of optical matter arrays of charged NPs whose structures are determined by both optical binding and electrostatic interactions. **c** Average nearest neighbor distances ($d_{avg}$) for the arrays with and without added charges. **d** Calculated optical torques on charged and uncharged arrays. Torque values for type-II and III arrays are also shown for comparison. Data points are mean values with one standard deviation error bars in both **c** and **d**

the optical binding lattice sites. No optical torque is observed in Fig. 5a while negative optical torque occurs in Fig. 5b when the interactions are turned on. Thus, it is the scattering from neighbors that induces (new) moments in the NPs that can collectively manifest negative torque and reversal.

We further consider the issues of symmetry and interparticle separation with intuitive theory and simulations. The results presented in Supplementary Note 10 make clear the importance of symmetry in the emergence of negative optical torque when the sign of the torque changes from twofold to threefold rotational symmetry. Comparison of results for calculations in the point-dipole approximation (treated only to first order scattering) to GMT and FDTD simulations show that the finite size of the objects also plays a key role in determining the (sign of the) torque on optical matter arrays.

The extended optical matter arrays can support collective excitations termed LP modes[23,29]. We find a close connection between the LP modes and the negative optical torque phenomenon. For trigonal lattices, the locations of the peaks of the LP modes can be approximated by the equation[29]

$$\lambda_{LP} = \Delta \left[ \frac{4}{3} \left( i^2 + ij + j^2 \right) \right]^{-\frac{1}{2}} \left( \frac{\varepsilon_{Ag}(\lambda_{LP})\varepsilon_b}{\varepsilon_{Ag}(\lambda_{LP}) + \varepsilon_b} \right)^{\frac{1}{2}} \quad (1)$$

where $\lambda_{LP}$ is the LP resonance wavelength, $\Delta$ is the lattice spacing, $\varepsilon_{Ag}(\lambda_{LP})$ is silver's permittivity function, $\varepsilon_b$ is the permittivity of the medium, and $i$ and $j$ are integers that enumerate different LP modes. This equation assumes a surface plasmon polariton dispersion, which is a more significant approximation in the case

of particle arrays as opposed to hole arrays for which it was proposed[29], but nonetheless leads to superior results than simply a grating condition that would not have the metal dielectric function term.

Figure 5c, d shows the total scattering and torque on 12-NP optical matter arrays with variable lattice spacing and incident wavelength. The resonance frequencies for the arrays are in very good agreement with those predicted by Eq. (1) (Supplementary Note 11). The LP modes have scattering peaks that closely coincide with zero-crossings in the optical torque, i.e., as one passes through resonance (a maximum in the scattering cross section) the sign of the torque changes (i.e., the torque goes through a zero). The correspondence was also observed for many other array sizes (i.e., with different number of NPs) we have simulated by the GMT method. This correspondence may be related to an effective polarizability of the system where the cross section is proportional to its absolute square whereas the largest contributions to the forces are related to its real part, with the latter having a zero at resonance. Similar correspondence between the reversal of optical torque and plasmonic resonance was also observed in nanoscale plasmonic motors[30].

The fact that the optical torque is positive on the blue side and negative on the red side of a LP resonance connects the spectroscopic properties of the NP cluster to its dynamical properties, and further clarifies the regions of negative torque observed in the torque phase diagram (Fig. 3). The crossover from positive to negative torque at lattice spacings around 647 nm for larger optical matter arrays coincides with the $ij = 01$ LP resonance peak. The emergence of a negative torque region at small lattice spacings for 8 or more particles is explained by the

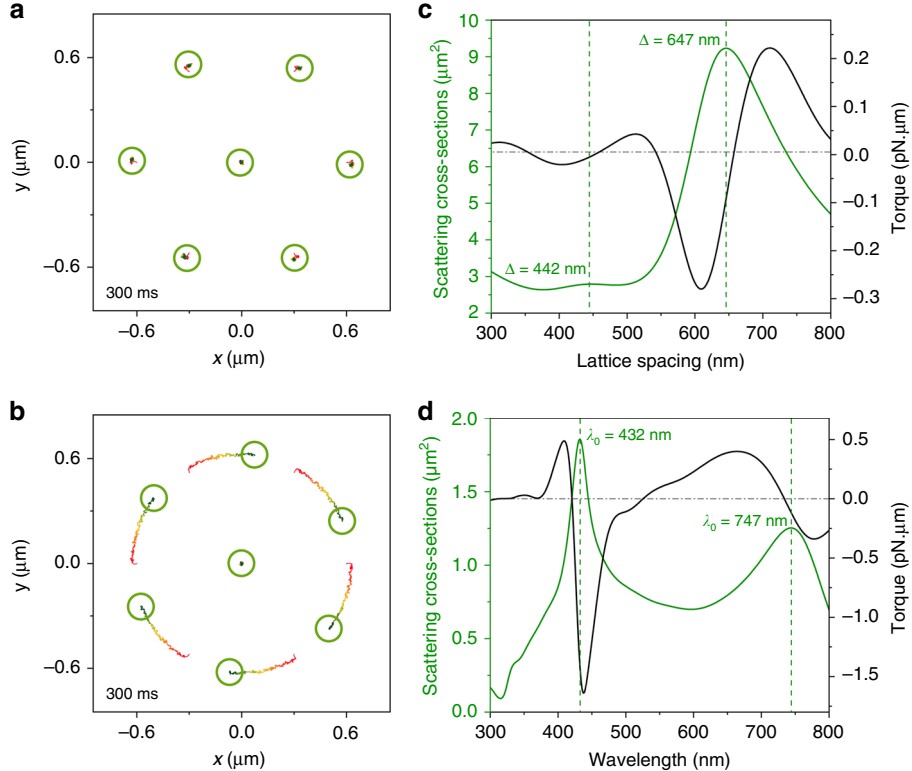

**Fig. 5** Optical interactions among NPs in optical matter arrays. **a** GMT-Langevin dynamics simulations of a 7-NP array where NP optical interactions are turned off. **b** The interactions are turned on causing rotation of the array. **c** Scattering cross-sections and optical torque for a 12 NP optical matter array built on an equilateral trigonal lattice with variable lattice spacing, while the illumination wavelength is fixed at 800 nm and in a medium with refractive index $n$ = 1.33. The scattering peak at $\Delta$ = 647 nm corresponds to the 01 LP mode predicted by Eq. (1). **d** Scattering cross-sections and optical torque of the same array for different incident wavelengths. The lattice spacing is fixed at 600 nm. The scattering peak at $\lambda$ = 747 nm corresponds to the 01 LP mode while the peak at $\lambda$ = 432 nm corresponds to the 11 mode

emergence of an LP mode that is insignificant in smaller optical matter arrays. The role of the LP mode in determining the sign of optical torque is also consistent with the experiments for $\lambda$ = 710 nm discussed above: Fig. 5d shows that this wavelength is to the blue of the LP resonance position which would lead to positive torque (as observed), whereas $\lambda$ = 800 nm incident light is to the red of the LP resonance position and in a region of negative torque.

## Discussion

Our theory and simulations agree with previous theoretical work[15]. Together, they demonstrate that the emergence of negative optical torque depends on both discrete rotational symmetry and phase retardation. Specifically, when a NP in an array is illuminated by light, the scattered photons exert a recoil torque on the NP. Negative optical torque occurs if the recoil angular momentum $-m\hbar$ is negative, where $m$ is the azimuthal component of the angular momentum index for the scattered wave and $\hbar$ is Planck's constant[15]. Chen et al. suggest that small clusters do not favor negative optical torque because they cannot access an azimuthal channel with high angular momentum (i.e., $m > 0$)[15], while larger clusters are more likely to generate negative optical torque. Our experimental observation of positive to negative optical torque crossover in optical matter arrays with increasing number of NPs supports this argument.

In addition, our discovery of the correspondence between negative optical torque and the lattice plasmon modes suggests that one can view an optical matter array as being a tunable electrodynamic material with an effective many-body (anisotropic) polarizability[17,31], which connects the optical matter

arrays to the anisotropic and inhomogeneous materials that exhibit negative optical torque due to spin–orbit optical interactions[8,16]. Optical matter, however, offers greater tunability of parameter space, including the wavelength of light, the interparticle separation, geometry and symmetry of the array, and the size, shape and material of the particles, making optical matter an intriguing system for exploring new optomechanical phenomena.

By organizing simple metal NPs into various mesoscale optical matter arrays via optical binding, the present study reveals a new approach to control and switch optical torque in optical materials. The optical matter arrays are reconfigurable, and therefore will allow programmable transduction of light into mechanical forces. The rich phenomena reported herein provide new opportunities for optomechanical, microrheological, and biological applications with directed conversion of photon momenta into optical torque with easily observable motions. In particular, optical torque switching in optical matter arrays could allow sensing the variation of fluid properties. The mesoscale optical matter arrays could serve as mechanical motors and rheological probes that can be assembled in situ in lab-on-a-chip devices[32] or even in cells by incorporating biocompatible NPs via endocytosis[33].

## Methods

**Experiments**. The Ag NPs purchased from NanoComposix (NanoXact™) are nearly spherical with diameters of 151 ± 13 nm and are coated by negatively charged polyvinylpyrrolidone with a zeta potential of –56 mV. A suspension of Ag NPs diluted in deionized water filled a chamber (8 mm diameter and 0.13 mm height) created with a spacer between two coverslips (Corning). The coverslips were treated by Hellmanex® III (0.5% by Vol) for 30 min before the experiments. All experiments were performed using an optical trapping system (Supplementary Figure 1) built with an inverted Olympus IX73 microscope and CW Tunable

Ti:Sapphire Laser (Spectra-Physics 3900 s) operating at 800 nm producing a TEM$_{00}$ Gaussian mode. The laser output was expanded and collimated to a beam with (1/$e^2$) diameter of 13.5 mm. The central area ($d = 2.7$ mm) of the beam was selected by an iris, then collimated and shrunk by a telescope with two lenses of $f = 75$ cm and 15 cm, and finally focused by a $f = 50$ cm lens to the back aperture of a 60X objective (NA = 1.2, Olympus UPLSAPO 60XW). The laser power was 91 mW after the iris. The objective created a focused laser spot of 5.5 μm diameter at the top coverslip to confine the NPs. The polarization state of the light was controlled by a quarter-wave plate. Trapped Ag NPs were visualized by darkfield microscopy (NA = 1.4, Olympus U-DCW condenser), and recorded using a CMOS camera (Point-Grey Grasshopper 3) with frame rate of 300 fps. Determination of particle positions in each frame and linking the particles in different frames were performed with TrackMate in ImageJ[34].

**FDTD simulations of static optical matter arrays**. The simulations were performed with Lumerical FDTD Solutions software. A circularly polarized plane wave with $\lambda = 800$ nm (vacuum) illuminated the Ag spheres (diameter 150 nm, refractive index $n = 0.04 + i5.57$) immersed in water ($n = 1.33$). The Ag spheres were located at the $z = 0$ plane and viewed from the negative $z$ position. The optical field propagated along the $+z$ direction, and its electric vector rotated counterclockwise at the $z = 0$ plane as perceived from the $-z$ point of view. Optical forces on individual Ag NPs were calculated using the Maxwell stress tensor method, and the optical torque on an optical matter array was obtained by calculating and summing the torque on individual Ag particles relative to the center of mass of the array. Optical torques in optical matter arrays were calculated by initially placing the particles on the lattice sites of a perfect hexagonal lattice with different lattice constants.

**FDTD-particle dynamics simulations**. The dynamic evolution of the NP system is enabled by predicting the next positions of all NPs driven by optical forces[35]. In the simulations, we first calculate the total force **F** for each NP contributed by optical forces and/or Coulomb forces at the current position, combined with the fluid drag force that obeys Stokes' law,

$$\frac{d\nu}{dt} = -\frac{\zeta}{m}\nu + \frac{\mathbf{F}}{m}, \tag{2}$$

where $m$ is the mass of a NP, $\nu$ is the velocity, and $\zeta = 6\pi\mu R$ is the friction coefficient ($\mu$ is the dynamic viscosity of water and $R$ is the radius of the NP). We assume the acceleration is negligible on the nanometer length scale and in the overdamped limit. By setting $d\nu/dt = 0$ in Eq. (2), we obtain the steady state velocity and calculate the next position

$$\mathbf{r}_{n+1} = \mathbf{r}_n + \nu_n dt. \tag{3}$$

In each simulation step, the particle movements are evaluated using a relatively large $dt$. If the maximum displacement $S_{max}$ for all NPs is larger than a specified value (set as 30 nm in our simulations), the $dt$ value is decreased and all displacements are recalculated until $S_{max} \leq 30$ nm. Optical forces on individual particles at the new positions are then calculated, and the simulation loop repeats to find the next positions. The simulations are performed with a customized code written in Lumerical FDTD Solutions software.

**GMT simulations**. The equation of motion for an N-particle system undergoing dissipation and thermal noise is given by the Langevin equation[36]

$$m\frac{d^2\mathbf{r}}{dt^2} = \mathbf{F}(\mathbf{r}, t) - \zeta\frac{d\mathbf{r}}{dt} + \boldsymbol{\eta}, \tag{4}$$

where **F** is the electrodynamic force on the particle and **η** is a Gaussian noise term with the usual properties for the mean, variance and noise so that the fluctuation-dissipation theorem holds. Eq. (4) is integrated in time using a leap-frog Verlet integrator to give the trajectories of the NPs.

The electrodynamic interactions and forces, **F**, for Eq. (4) are computed using the GMT method where in the incident and scattered fields are expanded into the vector spherical harmonic (VSH) functions for each particle[37,38]. The incident field is expanded into the regular VSH's $\mathbf{N}_{nm}^{(1)}$ and $\mathbf{M}_{nm}^{(1)}$,

$$\mathbf{E}_{inc}^j = -\sum_{n=1}^{L_{max}}\sum_{m=-n}^{n} iE_{mn}[p_{mn}^j\mathbf{N}_{nm}^{(1)} + q_{mn}^j\mathbf{M}_{mn}^{(1)}], \tag{5}$$

where $L_{max}$ is the maximum number of multipole orders to expand in, $\mathbf{E}_{mn}$ is a normalization constant, and $p_{mn}$ and $q_{mn}$ are the expansion coefficients to be solved for. The scattered field is expanded into the scattering VSH's $\mathbf{N}_{nm}^{(3)}$ and $\mathbf{N}_{mn}^{(3)}$,

$$\mathbf{E}_{scatt}^j = -\sum_{n=1}^{L_{max}}\sum_{m=-n}^{n} iE_{mn}\left[a_n^j p_{mn}^j\mathbf{N}_{nm}^{(3)} + b_n^j q_{mn}^j\mathbf{M}_{mn}^{(3)}\right], \tag{6}$$

where $a_n^j$ and $b_n^j$ are the ordinary Mie coefficients[39] of particle $j$.

The expansion coefficients are solved for in a system of $2N L_{max}(L_{max} + 2)$ equations,

$$p_{mn}^j = p_{mn}^{(j\rightarrow j)} - \sum_{l\neq j}^{(1,N)}\sum_{v=1}^{L_{max}}\sum_{u=-v}^{v} A_{mn}^{uv}(l \rightarrow j)a_v^l p_{uv}^l + B_{mn}^{uv}(l \rightarrow j)b_v^l q_{uv}^l,$$

$$q_{mn}^j = q_{mn}^{(j\rightarrow j)} - \sum_{l\neq j}^{(1,N)}\sum_{v=1}^{L_{max}}\sum_{u=-v}^{v} B_{mn}^{uv}(l \rightarrow j)a_v^l p_{uv}^l + A_{mn}^{uv}(l \rightarrow j)b_v^l q_{uv}^l, \tag{7}$$

where $p^{(j\rightarrow j)}$ and $q^{(j\rightarrow j)}$ are the expansion coefficients of the incident source and $A_{mn}^{uv}(l \rightarrow j)$ and $B_{mn}^{uv}(l \rightarrow j)$ are VSH translation coefficients from particle $i$ to particle $j$. Solving this system includes induced dipole interactions, as well as many-body interaction terms.

To turn off particle interactions, we simply set the expansion coefficients to the incident expansion coefficients instead of solving Eq. (7),

$$p_{mn}^j = p_{mn}^{(j\rightarrow j)},$$
$$q_{mn}^j = q_{mn}^{(j\rightarrow j)}. \tag{8}$$

Once the expansion coefficients are solved for, the force on each particle can be determined by integrating the Maxwell stress tensor **T** over the surface of each sphere,

$$\mathbf{F} = \oint_{\Omega} \mathbf{T} \cdot d\Omega. \tag{9}$$

These forces are then fed into the Langevin equation of motion, Eq. (4).

**Code availability**. The code for FDTD-particle dynamics and GMT simulations is available from the corresponding author upon reasonable request.

## Data availability

All relevant data used in this paper are available from the corresponding author upon reasonable request.

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

## Acknowledgements

We acknowledge support from the W. M. Keck Foundation Research Program. We also acknowledge partial support from the Vannevar Bush Faculty Fellowship program sponsored by the Basic Research Office of the Assistant Secretary of Defense for Research and Engineering and funded by the Office of Naval Research through grant N00014-16-1-2502. The FDTD-particle dynamics simulations are based upon work supported by the National Science Foundation under Grant No. 1610271. The GMT simulations were performed at the University of Chicago Research Computing Center under an award for computing time to N.F.S and J.A.P. This work was performed, in part, at the Center for Nanoscale Materials, a U.S. Department of Energy Office of Science User Facility, and supported by the U.S. Department of Energy, Office of Science, under Contract No. DE-AC02-06CH11357.

## Author contributions

Z.Y. and N.F.S. conceived of and Z.Y. designed the experiments. F.H. performed the experiments and F.H. and Z.Y. analyzed the data. Y.Y. performed additional independent measurements to verify the negative optical torque. Z.Y. performed all FDTD simulations. J.A.P., S.K.G and N.F.S. performed the GMT simulations. C.P. and N.F.S. performed calculations in the point-dipole approximation. Z.Y., N.F.S., S.K.G. and J.A.P. wrote the manuscript. All authors discussed the results and commented on the manuscript.

## Additional information

**Competing interests:** The authors declare no competing interests.

