## [Peer Review File · Nature Communications]

Reviewers' Comments:

Reviewer #1:

Remarks to the Author:

The paper is well written, easy to follow and the authors paid extraordinary attention to many aspects of experimental and theoretical procedures. I find rotation of optically bound structures self-arranged from nanoparticles very exciting and novel. Even though the experimental system has been used by the authors for several years doing experiments with self-arrangement of nanoparticles (see references in the paper), the submitted manuscript provides concise (in the particular geometry) description of the influence of optically bound particles on the rotation of the whole optically bound structure. Except precise analyses of the experimental data the authors also presented extended theoretical simulations of the optical self-arrangement, including also several other interfering mechanisms (reflection from the surface, thermophoresis, charged particles).

I suggest to publish the paper as it is, however with added x-label to Fig. S10b.

Reviewer #2:

Remarks to the Author:

In my opinion, the manuscript reported two major observations.

(1) Negative rotation of metallic nanoparticle clusters.

(2) Self-organization of metallic particles into rotating states.

In addition to a series of systematic experiment, comprehensive calculations are also given.

I would recommend the publication of the manuscript for one reason: to my knowledge, this is a first experimental demonstration of the phenomenon. It would highly relevant and beneficial if the author can dig out the principles of the phenomenon, such as when the torque is negative, what factor favors the negative torque and stable particle binding, etc. However, as someone who also has been working in similar problems, it is understood that it is unlikely to find an elegant and simple principles, as everything are obscured by the complex multiple scattering of light.

Here are some of the issues that the authors should consider.

(1) The underlining physical principles of negative rotation is not clearly stated in the manuscript.

These principles can be found elsewhere in the literature, such as ref. 17 in the manuscript. I believe it would be of interest to the reader if the authors could add a paragraph about the underlining principles.

(2) Any torque in the system can only be due to spin-orbital angular momentum conversion. It is known that small cluster, owing to its physical size, cannot access azimuthal channel with high angular momentum, and therefore cannot be entitled to negative rotation. Is this what is happening in the experiment?

(3) How the particles are stabilized in the longitudinal direction? Are they pressed against a glass surface? Does the reflection interfere to produce some effect?

Reviewer #3:

Remarks to the Author:

This work presents angular optomechanics of optically-bounded clusters of nanoparticles, richly supported by thorough experimental and simulation data. The main point of the study is to unravel the appearance of co-rotating or contra-rotating motion of 2D clusters of nanoparticles with respect to the incident spin angular momentum. This leads the authors to make an emphasis on "negative" torque, which corresponds to the contra-rotating case. In overall, the study is properly conducted and discussed. The topic is interesting and timely and should attract the interest of optics &

photonics community. In overall, I am therefore in favor of its publication in Nature Communications, however, a few concerns should be addressed before:

- abstract: negative torque is said to have the potential for several applications, however, it should be noted that positive torque as well... from practical mechanical point of view co- or contra-rotation can also be simply actuated by changing the incident handedness of the circular polarization state. What seems the most interesting is the reconfigurability of the clusters that may have different mechanical responses. Rephrasing is welcome, in particular I think that deleting “negative” from the last sentence would give a broader picture.

- introduction (and abstract): I am not really convinced by the opposition made by the authors regarding non-conservative mechanical effects and conservation laws. I feel that this could be misleading to make such an emphasis.

- p2: negative torque theoretically postulated in refs. 16,17 (from 2009 and 2014): it is worth considering the paper J. Opt. Soc. Am. A 24, 430 (2007) dealing with negative torque on spheroidal microparticles. Regarding experimental realization, a very recent one Nat. Photon. 12, 461 (2018) dealing with steady observations at the macroscale would complement the state-of-the-art.

- p5, section “crossover...”: “total orbital torque exerted on the whole array”: as such, one can speak about a torque inducing spinning of the cluster. I think “orbital” qualification is unclear. - p5, next sentence: incident spin angular momentum (SAM) is said to be converted into orbital motion: I do not agree with such a statement. Indeed, SAM is deposited via expected polarization changes (though not discussed or reported) while light scattering and corresponding linear momentum distribution implies orbital angular momentum (OAM) transfer to the cluster. Still, SAM and OAM are two inter-dependent channels for optical angular momentum, the interplay being controlled by the interaction of light with the cluster.

- p5, next sentence: individual spinning is expected, but said undetectable because of spherical symmetry, while later in the paper the non-spherical symmetry of actual particles is discussed... By the way, may be temporal Fourier spectra of transmitted light could reveal spinning motion.

- p5, next sentence: SAM does not contribute to the orbital torque: here again there is the issue of “orbital torque” mentioned above. However, there is another concern: coupled spinners may collectively orbit, even in absence of SAM-OAM coupling, for instance via hydrodynamic interaction.

- Did the authors performed experiments with incident linear polarization? Indeed, the wavelength study and positive-to-negative rotation recalls the following study: Nat. Nano. 5, 570 (2010) where transition between co/contra rotations of structured nanoparticles exhibiting plasmonic resonances were observed and discussed as the wavelength changes.

- p9, Discussion, end of first paragraph: “converts SAM to OAM” see concerns above

- p9, next sentence: rotational symmetry and phase retardation effects are pointed out in the framework of nanoparticles. Since authors mention effective anisotropic polarizability of intrinsically isotropic structures, the latter connection could be most probably by fruitfully extended to the case of anisotropic materials whose rotational symmetries are closely linked to positive/negative torques (ref.7).

p11, connection between resonance and co/contra rotational motion: there is no one-to-one correspondence, as one can see from Figs. 5c and 5d: could the residual "misconnection" be due to small effects like thermal ones discussed in the SI?

Response

We thank the editor and reviewers for the time they have taken to carefully examine our manuscript (NCOMMS-18-20437). The reviewers' thoughtful comments helped us focus our revisions on issues that were unclear and thus have strengthened the revised manuscript. We have revised the manuscript in response to all the comments, as detailed in the point-by-point response below. These revisions are indicated in red color in the revised manuscript.

Reviewer 1

The paper is well written, easy to follow and the authors paid extraordinary attention to many aspects of experimental and theoretical procedures. I find rotation of optically bound structures self-arranged from nanoparticles very exciting and novel. Even though the experimental system has been used by the authors for several years doing experiments with self-arrangement of nanoparticles (see references in the paper), the submitted manuscript provides concise (in the particular geometry) description of the influence of optically bound particles on the rotation of the whole optically bound structure. Except precise analyses of the experimental data the authors also presented extended theoretical simulations of the optical self-arrangement, including also several other interfering mechanisms (reflection from the surface, thermophoresis, charged particles).

I suggest to publish the paper as it is, however with added x-label to Fig. S10b.

Reply: We thank the reviewer for the supportive remarks and careful examination of our manuscript. We have added “ d (nm)” as the x-label for Fig. S10b (now as Fig. S11b).

Reviewer 2

In my opinion, the manuscript reported two major observations.

- (1) Negative rotation of metallic nanoparticle clusters.
- (2) Self-organization of metallic particles into rotating states.

In addition to a series of systematic experiment, comprehensive calculations are also given. I would recommend the publication of the manuscript for one reason: to my knowledge, this is a first experimental demonstration of the phenomenon. It would highly relevant and beneficial if the author can dig out the principles of the phenomenon, such as when the torque is negative, what factor favors the negative torque and stable particle binding, etc. However, as someone who also has been working in similar problems, it is understood that it is unlikely to find an elegant and simple principles, as everything are obscured by the complex multiple scattering of light.

Comments 1: *The underlining physical principles of negative rotation is not clearly stated in the manuscript. These principles can be found elsewhere in the literature, such as ref. 17 in the manuscript. I believe it would be of interest to the reader if the authors could add a paragraph about the underlining principles.*

Reply 1: Thank you for the suggestion. We have added a new paragraph to discuss the physical principles as quoted below.

Added in main text (Page 12): Our theory and simulations agree with the previous theoretical work¹⁵, which has shown that the emergence of negative optical torque depends on both discrete rotational symmetry and phase retardation. Specifically, when a NP in an array is illuminated by light, the scattered photons exert a recoil torque on the NP. Negative optical torque occurs if the recoil angular momentum $-m\hbar$ is negative, where m is the azimuthal component of angular momentum index for the scattered wave and \hbar is the Planck constant¹⁵. Chen *et al.* suggest that small clusters do not favor negative optical torque, because they cannot access an azimuthal channel with high angular momentum (*i.e.*, $m > 0$)¹⁵, while larger clusters are more likely to generate negative optical torque. Our experimental observation of positive to negative transition of optical torque in optical matter arrays with increasing number of NPs supports this argument. In addition, our discovery of the correlation between negative optical torque and the lattice plasmon modes suggests that one can view an optical matter array as being a tunable electrodynamic material with an effective many-body (anisotropic) polarizability^{17,31}, which connects the optical matter arrays to the anisotropic and inhomogeneous materials that have shown negative optical torque due to spin-orbit optical interactions^{8,16}. Optical matter, however, offers greater tunability of parameter space, including the wavelength of light, the interparticle separation, geometry and symmetry of the array, and the size, shape and material of the particles, making optical matter an intriguing system for exploring new optomechanical phenomena.

Comments 2: *Any torque in the system can only be due to spin-orbital angular momentum conversion. It is known that small cluster, owing to its physical size, cannot access azimuthal channel with high angular momentum, and therefore cannot be entitled to negative rotation. Is this what is happening in the experiment?*

Reply 2: We agree with this comment, and note that this has been predicted by Chen *et al.* in Ref. 15 (Ref. 17 in the previous manuscript). We have added some discussion in the new paragraph (see above).

Comments 3: *How the particles are stabilized in the longitudinal direction? Are they pressed against a glass surface? Does the reflection interfere to produce some effect?*

Reply 3: In the experiments, the particles are pushed by optical scattering forces against the upper coverslip surface of a liquid sample cell. The particles are stabilized by a potential well (see Fig. S11b) created by the scattering force and electrostatic repulsive forces between the particles and glass surface. The reflection interference has little influence. Details on these factors have been discussed in Supplementary Note E.

Reviewer 3

This work presents angular opto-mechanics of optically-bounded clusters of nanoparticles, richly supported by thorough experimental and simulation data. The main point of the study is to unravel the appearance of co-rotating or contra-rotating motion of 2D clusters of nanoparticles with respect to the incident spin angular momentum. This leads the authors to make an emphasis on “negative” torque, which corresponds to the contra-rotating case. In overall, the study is properly conducted and discussed. The topic is interesting and timely and should attract the interest of optics & photonics community. In overall, I am therefore in favor of its publication in Nature Communications, However, a few concerns should be addressed before.

Comments 1: *Abstract: negative torque is said to have the potential for several applications, however, it should be noted that positive torque as well... from practical mechanical point of view co- or contra-rotation can also be simply actuated by changing the incident handedness of the circular polarization state. What seems the most interesting is the reconfigurability of the clusters that may have different mechanical responses. Rephrasing is welcome, in particular I think that deleting “negative” from the last sentence would give a broader picture.*

Reply 1: That is a good suggestion. We have revised the abstract to emphasize both positive and negative optical torque.

Revised in the abstract: Our results establish that *both positive and negative optical torque can be readily realized and controlled in optical matter arrays, which are reconfigurable and can potentially lead to programmable materials for optomechanical, microrheological and biological applications.*

Comments 2: *introduction (and abstract): I am not really convinced by the opposition made by the authors regarding non-conservative mechanical effects and conservation laws. I feel that this could be misleading to make such an emphasis.*

Reply 2: We have removed the sentences related to non-conservative mechanical effects and conservation laws from the abstract and introduction.

Comments 3: *p2: negative torque theoretically postulated in refs. 16,17 (from 2009 and 2014): it is worth considering the paper J. Opt. Soc. Am. A 24, 430 (2007) dealing with negative torque on spheroidal microparticles. Regarding experimental realization, a very recent one Nat. Photon. 12, 461 (2018) dealing with steady observations at the macroscale would complement the state-of-the-art.*

Reply 3: Thank you for the suggestion. We have cited these papers in the revised manuscript.

Comments 4: *p5, section “crossover...”: “total orbital torque exerted on the whole array”: as such, one can speak about a torque inducing spinning of the cluster. I think “orbital” qualification is unclear.*

Reply 4: We have added a sentence in the main text that “Note here the negative torque refers to the total orbital torque exerted on the whole array, as such one can also view the whole array

as spinning. However, due to the discrete nature of the array, we prefer to term its motion as rotation to better describe the orbital dynamics of the NPs.”

Comments 5: *p5, next sentence: incident spin angular momentum (SAM) is said to be converted into orbital motion: I do not agree with such a statement. Indeed, SAM is deposited via expected polarization changes (though not discussed or reported) while light scattering and corresponding linear momentum distribution implies orbital angular momentum (OAM) transfer to the cluster. Still, SAM and OAM are two inter-dependent channels for optical angular momentum, the interplay being controlled by the interaction of light with the cluster.*

Reply 5: We have removed that sentence to avoid confusion.

Comments 6: *p5, next sentence: individual spinning is expected, but said undetectable because of spherical symmetry, while later in the paper the non-spherical symmetry of actual particles is discussed... By the way, may be temporal Fourier spectra of transmitted light could reveal spinning motion.*

Reply 6: We have changed the sentence to “We also expect that the individual NPs are spinning about their individual axes in circularly polarized light, *which is not directly observable in optical images but has been demonstrated by the oscillating autocorrelation function of scattering intensity*⁶.”

Comments 7: *p5, next sentence: SAM does not contribute to the orbital torque: here again there is the issue of “orbital torque” mentioned above. However, there is another concern: coupled spinners may collectively orbit, even in absence of SAM-OAM coupling, for instance via hydrodynamic interaction.*

Reply 7: We have examined the hydrodynamic coupling in our arrays, but the influence on orbiting direction is insignificant. We have added new sentences in the main text and a new section in the supplementary information.

Added in main text (Page 5): It is worth noting that hydrodynamic coupling may occur between two optically bound and spinning particles, where the liquid flow created by the orbiting particles may reverse their spin direction for certain particle sizes¹⁴. We have examined the hydrodynamic coupling in our optical matter arrays but did not find any change of orbiting direction caused by coupled spin rotation of the 150 nm Ag NPs.

Added new section in the supplementary information: Note B. Influence of the spin of single NPs to the rotation of optical matter arrays under hydrodynamic coupling

The spin angular momentum of the incident light couples into the spinning of the individual NPs (primarily through absorption). We have examined the influence of the spin of single NPs to the rotation of the entire NP arrays under hydrodynamic coupling, and the calculation results are shown in Fig. S7. The electrodynamic spin torque is generally positive, even when the orbital torque on the cluster is negative and is about 50 times weaker than the orbital torque. The spinning NPs inside a fluid then leads to a hydrodynamic coupling in addition to the electrodynamic coupling. However, this hydrodynamic coupling cannot be an explanation for the observed negative orbital torque since the NP spinning is always a positive torque and will therefore lead to a positive contribution to the orbital motion through hydrodynamic coupling. There is no retardation length scale for the hydrodynamic interactions in this system, so the

orbital torque cannot switch signs like it does in the electrodynamic coupling case. Thus, the experimental observation of negative orbital torque is made despite the presence of hydrodynamic coupling interactions working against the electrodynamic interactions.

Fig. S7 | Spin and orbital torque as a functions of the lattice spacing in a 7-NP array.

The dashed arrow in the inset indicates the rotation direction of the electric vector of light, and the solid arrows indicate the rotation directions of individual NPs or the whole array.

Comments 8: *Did the authors performed experiments with incident linear polarization? Indeed, the wavelength study and positive-to-negative rotation recalls the following study: Nat. Nano. 5,570 (2010) where transition between co/contra rotations of structured nanoparticles exhibiting plasmonic resonances were observed and discussed as the wavelength changes.*

Reply 8: Yes, we also performed experiments with incident linear polarization (for example, see McCormack, P., Han, F. & Yan, Z. *J. Phys. Chem. Lett.* 9, 545-549 (2018)). The silver nanoparticles will form stationary clusters without rotation under linear polarization. We have added a sentence in the main text and cited the Nat. Nano. paper.

Added in main text (Page 11): Similar correlation between the reversal of optical torque and plasmonic resonance was also observed in nanoscale plasmonic motors³⁰.

Comments 9: *p9, Discussion, end of first paragraph: “converts SAM to OAM” see concerns above.*

Reply 9: We have removed “converts SAM to OAM”.

Comments 10: *p9, next sentence: rotational symmetry and phase retardation effects are pointed out in the framework of nanoparticles. Since authors mention effective anisotropic polarizability of intrinsically isotropic structures, the latter connection could be most probably by fruitfully extended to the case of anisotropic materials whose rotational symmetries are closely linked to positive/negative torques (ref.7).*

Reply 10: Thanks for the thoughtful suggestion. We have added some discussions in the main

text.

Added in main text (Page 12): In addition, our discovery of the correlation between negative optical torque and the lattice plasmon modes suggests that one can view an optical matter array as being a tunable electrodynamic material with an effective many-body (anisotropic) polarizability^{17,31}, *which connects the optical matter arrays to the anisotropic and inhomogeneous materials that have shown negative optical torque due to spin-orbit optical interactions*^{8,16}. *The optical matter, however, offers greater tunability of parameter space, including the wavelength of light, the interparticle separation, geometry and symmetry of the array, and the size, shape and material of the particles, making the optical matter an intriguing system for exploring new optomechanical phenomena.*

Comments 11: *p11, connection between resonance and co/contra rotational motion: there is no one-to-one correspondence, as one can see from Figs. 5c and 5d: could the residual “misconnection” be due to small effects like thermal ones discussed in the SI?*

Reply 11: **Actually, the correlation is between zeros in the optical torque or positive/negative crossings and the resonance positions.** This correlation is not absolutely perfect but reasonably good. We have examined many other array sizes and they all show the same correlation, and the residual “misconnection” is unlikely due to thermal effects since the calculations are based on pure electrostatics. We have added some discussion in the main text.

Added in main text (Page 11): The resonant frequencies for the arrays are in very good agreement with those predicted by Eq. 1 (Supplementary Note K), *and the LP modes have scattering peaks that closely coincide with zero-crossings in the optical torque, i.e., as one passes through resonance (a maximum in the scattering cross section) the sign of the torque changes (the torque goes through a zero). The correlation is not absolutely perfect but reasonably good and has been observed in numerous other array sizes we have carried out calculations for. This correlation may be related to an effective polarizability of the system such that the cross section is proportionate to its absolute square whereas the largest contributions to the forces are related to its real part, with the latter having a zero at resonance.*

Again, we appreciate the detailed comments and suggestions from the reviewers.

REVIEWERS' COMMENTS:

Reviewer #1 (Remarks to the Author):

The authors address the reviewers' comments and I suggest to accept the manuscript.

Reviewer #2 (Remarks to the Author):

The authors have taken my concerns into account in this round of review. I recommend its publication in Nature Communications.

Reviewer #3 (Remarks to the Author):

I carefully read the replies of authors to all the comments raised in the 3 reports. Provided answers and corresponding update of the manuscript make it better. I do recommend the publication of this revised version of the work.

Reviewers' Comments:

Reviewer #1 (Remarks to the Author):

The authors address the reviewers' comments and I suggest to accept the manuscript.

Reviewer #2 (Remarks to the Author):

The authors have taken my concerns into account in this round of review. I recommend its publication in Nature Communications.

Reviewer #3 (Remarks to the Author):

I carefully read the replies of authors to all the comments raised in the 3 reports. Provided answers and corresponding update of the manuscript make it better. I do recommend the publication of this revised version of the work.

**

Response

We thank the reviewers for reviewing and recommending the publication of our revised manuscript!